# Nonverbal communication in selfies posted on *Instagram*: Another look at the effect of gender on vertical camera angle

**Alessandro Soranzo** [1] *, **Nicola Bruno** [2]

**1** Sheffield Hallam University, Sheffield, United Kingdom, **2** Università di Parma, Parma, Italy

* a.soranzo@shu.ac.uk

**Data Availability Statement:** All data files are available from the osf database (accession number https://osf.io/qf8n3/).

**Funding:** The authors received no specific funding for this work.

## Abstract

### Background

Selfies are a novel social phenomenon that is gradually beginning to receive attention within the cognitive sciences. Several studies have documented biases that may be related to non-verbal communicative intentions. For instance, in selfies posted on the dating platform *Tinder* males but not females prefer camera views from below (Sedgewick, Flath & Elias, 2017). We re-examined this study to assess whether this bias is confined to dating selection contexts and to compare variability between individuals and between genders.

### Methods

Three raters evaluated vertical camera position in 2000 selfies– 1000 by males and 1000 by females—posted in *Instagram*.

### Results

We found that the choices of camera angle do seem to vary depending on the context under which the selfies were uploaded. On *Tinder*, females appear more likely to choose neutral, frontal presentations than they do on *Instagram*, whereas males on *Tinder* appear more likely to opt for camera angles from below than on *Instagram*.

### Conclusions

This result confirms that the composition of selfies is constrained by factors affecting non-verbal communicative intentions.

## Introduction

With the advent of smartphones equipped with front cameras and preview screens, many of us are increasingly taking photographic self-portraits, or "selfies", for purposes ranging from the social to the professional. Given their tremendous reach, popularity, and potential interest as a

**Competing interests:** The authors have declared
that no competing interests exist.

brand-new social phenomenon, selfies are gradually beginning to receive attention within the
cognitive sciences [1].

Among potentially fruitful approaches, especially interesting have been attempts to relate
compositional features of selfie images to communicative intentions of selfie-takers. Underly-
ing these attempts is the common idea that selfies may be studied as interactive objects which
may be interesting not only for their potential aesthetic value [2] but also for their role in social
exchanges [3]. Selfies are a novel form of non-verbal communication: by choosing specific
poses or by manipulating other pictorial features in the images, selfie-takers are assumed to
provide non-verbal social and emotional signals to potential viewers [4]. In a recent paper,
Bruno, Uccelli, Pisu, Belluardo & De Stefani [5] proposed that these signals may be conceptual-
ized, within the two dimensional framework of the picture, as analogous to the non-verbal sig-
nals that we use in face-to-face communication. In face-to- face interactions, individuals
control interpersonal distance, body posture, and facial expressions to modulate the quality of
the exchange on dimensions such as approach-avoidance, intimacy- social distance, or posi-
tive-negative affect. These behaviours are believed to have roots in territorial behaviours by
non-human animals [6, 7] and have been codified in detail by Hall [8] who grouped them
under the label of "proxemics." Proxemic space-related behaviours have been investigated
extensively [9]. For instance, classic [10] and recent [11, 12] evidence indicates that manipula-
tions of proxemic variables produce measurable psychophysiological responses within specific
brain structures. In addition, psychological effects of proxemic manipulations have been
shown to generalize to the digital context [13]. In selfies, space manipulations pertain only to
the composition of the image, such that an actual interpersonal distance cannot be defined.
The picture space, however, provides a reference frame for classifying and measuring different
space- related variables such as orientations, left-right asymmetries, and relative sizes. Such
proxemic features of pictures have been analysed, for instance, in relation to cinematic tech-
niques for placing characters with respect to the camera [14].

Taken together, the above considerations suggest that picture-related proxemic variables
may concur in defining the pose of the selfie-taker with respect with the implied viewpoint,
providing indications about the distance of the subject from the camera, the elevation of the
viewpoint below or above the subject, and its right-left position. These "selfie-specific" proxe-
mic cues may serve the purpose of non-verbally communicating the selfie-taker motivations,
intentions, or emotional states (see, [15]). Several studies [16–20] have documented a bias for
presenting the left over the right cheek to the camera when posing for a selfie. This bias has
been related to the communication of emotions under the control of the right cerebral hemi-
sphere [21, 22], in analogy with similar interpretations for painted portraits and self-portraits
[23, 24]. Other studies (Bruno, Abbati & Lamberti, in preparation; [25]) have shown that selfies
by females tend to have smaller facial prominence (the ratio of the extent occupied by the face
to that occupied by the body) than those of males. This sex-related bias has been related to gen-
der-stereotyped communication of one's self-representation, again in analogy with analogous
reports for photographic portraits in media and advertising [26] and online profile pictures
[27]. Our focus here is however on one last piece of evidence recently reported by Sedgewick,
Flath & Elias [28].

In their recent paper, Sedgewick and collaborators assessed the apparent vertical angle of
the camera in a sample of selfies posted on the online dating application *Tinder*, finding that
female takers were more likely to choose poses suggesting a higher vertical camera position
than male takers. Taking note that the selfies were posted on an online dating application, they
suggested that camera angle manipulations could be interpreted as means to provide cues for
mate selection. Specifically, Sedgewick and collaborators documented that males favour taking
selfies positioning the camera below their eye-level, whilst females prefer taking selfies from

above the eye-level. They suggested that selfies taken from below eye-level may suggest greater physical height and power, which are arguably features that may be attractive to females. Conversely, selfies taken from above, or at, eye-level may result in a slimmer figure, and selfies taken from above may induce tilting the head downwards. It has been suggested that these features tend be attractive to males [29–31].

Here we re-examine the finding of Sedgewick and collaborators asking two additional questions. First, is the sex-related difference in vertical camera angle confined to contexts that underscore mate selection, or does it generalize to other contexts of non-verbal communication? Second, how consistent are users in their choice of camera angle and how does their intra-individual variability compare to the variability that is observed between males and females? Both questions are relevant to how we might understand selfies as vehicles for nonverbal communication. To provide answers, we looked at samples of selfies publicly posted on the online photo-sharing application *Instagram*. In contrast to *Tinder*, this application encompasses a large variety of communicative intentions and content. For this reason, it represents a perfect source for testing whether effects generalize beyond a dating context. In addition, given that most users typically post multiple selfies, this choice also provided us with an opportunity to evaluate interindividual consistency in the choice of vertical camera angle. A better understanding of the potential role of selfies in interpersonal communication has theoretical and practial implications. At the theoretical level, assessments of the association between communicative intent and compositional choices represent a test of the predictions of current theories of non-verbal communication in selfies [5]. Practically, such assessment may provide useful in designing systems for automatically extracting user information from posted self portraits [32, 33], for applications aimed at health-related behavior modification [34–36], and for medical applications implementing internet-based patient monitoring [37–39].

## Materials and methods

Informed consent was waived by the ethics committee.

### Sample

A sample of 2000 selfies was collected from the photo-sharing media application *Instagram*. This figure was chosen based on Sedgewick and collaborators [28]'s sample of 962 selfies and an average effect size from their analysis was 0.44. We doubled this figure considering that whilst Sedgewick at al had independent selfies our sample was made by 10 selfies from the same users. Specifically, the selfies were garnered by a research assistant by searching with the #selfie hashtag on the application and recording the first 10 selfies from each of 100 male and 100 female users that had posted at least 10 selfies in their profile pages. The users' sex was determined by their first names as returned by the search engine. To test for the consistency of the camera angle, our sample included only selfie takers with at least 10 selfies. Selfies were defined as individual photographic self-portraits by the user, taken by a smartphone camera at arm's length. Thus, we discarded all self-portraits which included other individuals (sometimes called *groupies*, *wefies*, or *usies)* as well as photographic portraits that could not be selfies based on their content (for instance, if both hands of the sitter were visible in the picture). Similarly, we discarded selfies taken by capturing one's image in front of a mirror, as these have clearly different constraints on composition than standard selfies taken by looking at oneself in the smartphone preview screen.

### Ethics

User ID's and posting dates were recorded temporarily for the sole purpose of giving each selfie a unique identifier and then discarded. In adherence with the *Instagram* terms and

conditions, which prevents the distribution of user images without credit and consent, the selfies were stored privately and anonymously, they were analysed for the sole purpose of the current study and were not distributed or shared for any other purpose. The study was performed in accordance with the ethical standards of the Code of Ethical Principles for Medical Research Involving Human Subjects of the World Medical Association (Declaration of Helsinki), with the ethical standards of the Italian Board of Psychologists (see http:// www.psy.it/ codice_deontologico.html), as well as the Ethical Code for Psychological Research of the Italian Psychological Society (see http://www.aipass.org/node/26). The study was approved by the Sheffield Hallam university ethics panel (nr. E25008132/2019).

## Database and dependent variable

A database of 2000 cases was constructed by collating the selfie data for each of the 10 images per taker. For each selfie, we recorded: a unique number identifying the user, a number from 1 to 10 identifying the selfie, the gender of the user as apparent from the photograph (only two gender categories were recorded as we did not find selfies that suggested androgynous or genderqueer individuals), and our dependent variable which was a trichotomous, mutually exclusive classification of the vertical camera angle suggested by the selfie in question: "camera above", "camera below", and "camera frontal". Each selfie was classified as belonging to one of the three categories by three independent raters. For the large majority of the selfies the rating was straightforward, and all raters entered the same response; that is, there was an almost perfect overlap among raters. However, for 16 selfies (0.8%) the classification differed in at least one rater. These inconsistencies were resolved by retrieving the relevant images and by drawing three horizontal lines across the face, one through the bottom of the face, one through the eyes, and one at the top of the forehead. The orientation of the pose in relation to the position of the camera was then determined by comparing the horizontal extents between these lines. For example, if the distance between the lines on the forehead and on the eyes was larger than that between the lines on the eyes and the bottom of the face, the selfie was classified as taken from above eye level.

## Results

### Intra-Individual consistency

Table 1 presents the frequencies of rated vertical camera angles for male and female selfies for each selfie. The table documents an overabundance of female selfies classified as "camera above", and of male selfies classified as "camera frontal", whereas both females and males equally tended to avoid taking selfies from below.

Data were firstly analysed through a multimodal mixed logit model implemented in the mlogit package [40] in R version 3.6.1 [41] with camera vertical angle (above, below, or frontal) as the dependent variable, sex as the predictor variable, and participant as the random factor. This model revealed a strong overall effect of the predictor [$\chi^2(4) = 53.47$, p < 0.0001; McFadden $r^2$: 0.013].

**Table 1. Frequencies of rated camera height in female and male selfies.**

|  | Below | Frontal | Above | Total |
|---|---|---|---|---|
| **Female** | 184 | 323 | 493 | 1000 |
| **Male** | 208 | 455 | 337 | 1000 |
| **Total** | 392 | 778 | 830 | 2000 |

The model shows that females take significantly more selfies from above (coeff = 0.72; 95% CI [0.52, 0.92]; z = 7.13; p < 0.0001) whilst males take significantly more selfies from the frontal position (coeff = 0.5; 95% CI [0.92,0.52]; z = 4.07; p < 0.0001). The model did not show an important difference between the sexes for the below position (coeff = 0.22; 95% CI [-0.02, 0.46]; z = 1.77; p = 0.078).

## Interindividual consistency

To assess interindividual consistency in camera height choices, for each sampled taker we assessed whether the most frequent, or dominant, pose was from below, above, or frontal. Given that we sampled 10 selfies from each taker and that there are three possible poses, a pose is dominant when chosen for at least 4 selfies (4 vs 3 and 3), or for as many as 10. Although possible in principle, we did not encounter instances of an *ex-aequo* consisting of 4 and 4 vs 2 poses. Table 2 presents the frequencies of the dominant pose per sex. As one would expect given Tables 1 and 2 documents an overabundance of female selfies whose dominant pose was classified in the "above" category, of male selfies whose dominant pose is in the "frontal" category, whilst "below" is the least dominant pose for both males and females. Accordingly, a test of association confirms that these frequencies are not consistent with the null hypothesis of independence of sex and dominant pose category [$\chi^2(2) = 13.31$, p < 0.002; Cramer's V = 0.26].

The number of selfies associated to the dominant pose in each taker represents an index of inter-individual consistency for that taker. Table 3 presents the frequency distribution of these indices. In out of 200 selfie takers, 129 (64.5%) dominant poses involved at least 6 of the 10 selfies, and 83 (41,5%) at least 7. We take this as evidence that takers were fairly consistent in their choice of camera height, although only 8 takers chose the same angle in all selfies.

To evaluate if consistency in the choice of camera angles was comparable in males and females, we tabulated the data as a function of sex and number of dominant poses (Table 4) and ran a Bayesian test of association using version 0.9.12–4.2 of the BayesFactor package [42] using default priors and independent multinomial sampling plan. The resulting Bayes factor01 of 154.2 to 1 in favour of the null hypothesis indicates that males and females were similarly consistent in their preferences for a camera angle in the selfies.

## Comparison between platforms

A multimodal mixed logit model was also used to compare the data between the Instagram and Tinder platform. Tinder data were retrieved from Table 1 in Sedgewick et al ([28]; page 3). Fig 1 shows a direct comparison between the two projects.

We used camera vertical angle (above, below, or frontal) as the dependent variable; whilst sex, platform and their interaction were the predictor variables. This model revealed a strong overall effect of the predictors [$\chi^2(6) = 210.12$, p < 0.0001; McFadden r2: 0.038].

The model shows that, overall, females take significantly more selfies from above than males (coeff = 0.72; 95% CI [0.52, 0.9]; z = 7.13; p < 0.0001) whilst there was a tendency for males to take more selfies from the frontal position (coeff = 0.22; 95% CI [-0.02, 0.46]; z = 1.77; p = 0.08). Most interestingly, the model shows a strong platform difference: compared to

**Table 2. Frequencies of individual dominant poses in female and male selfies.**

|        | Below | Frontal | Above | Total |
|--------|-------|---------|-------|-------|
| Female | 15    | 28      | 57    | 100   |
| Male   | 12    | 53      | 35    | 100   |
| Total  | 27    | 81      | 92    | 200   |

**Table 3. Frequency distribution of number of dominant poses.**

| Dominance | 4 | 5 | 6 | 7 | 8 | 9 | 10 |
|-----------|---|---|---|---|---|---|----|
| N. Selfies | 21 | 50 | 46 | 34 | 28 | 13 | 8 |

Tinder, on Instagram there were significantly more selfies from above than from frontal (coeff = 0.78; 95% CI [0.61, 0.94]; z = 9.03; p < 0.0001) and slightly more selfies from frontal than from below (coeff = 0.51; 95% CI [0.091, 0.94]; z = 2.38; p < 0.05). The interactions show that for females there were significantly more selfies from above on Instagram than on Tinder (coeff = 1.67; 95% CI [1.34, 1.99]; z = 10.11; p < 0.0001) whilst there were a similar number of selfies from the other two vertical positions (p > 0.05).

For males, instead, there were significantly more selfies from below on Tinder than on Instagram (coeff = 0.94; 95% CI [0.60, 1.28]; z = 5.42; p<0.0001) and slightly less selfies from above (coeff = -.51; 95% CI [-0.93, -0.091]; z = -2.39; p = 0.02).

## Discussion

Our outcomes reproduced some of the features reported by Sedgewick and collaborators [28] on the *Tinder* data. However, our pattern of results also differed from the *Tinder* ones in non-trivial ways. To see how, consider again Fig 1. As can be seen in the figure, the two features that are clearly similar in the two datasets are that males prefer a frontal camera angle over the other two, and that females prefer the camera above angle more than males. The two datasets differ however in two other details. First of all, female selfies are much more likely to display a frontal angle on *Tinder* than they are on *Instagram*. Second, on *Tinder* male selfies are more likely to display an angle from below than female, whereas on *Instagram* the opposite seems to be the case: males avoid angles from below as females do.

The most interesting general conclusion stemming from this comparison is that choices of camera angle do seem to vary depending on the context under which the selfies were uploaded. On *Tinder*, females appear more likely to choose neutral, frontal presentations than they do on *Instagram*, whereas males on *Tinder* appear more likely to opt for camera angles from below than on *Instagram*. Our assessment of interindividual consistency further corroborates this conclusion, suggesting that female and males are similarly consistent in their choices. This pattern of results is therefore consistent with the idea that pictorial features such as camera angle are at least in part modulated by communicative intentions as expected if the composition of selfies were constrained by factors affecting non-verbal communication. As suggested by Sedgewick and collaborators, male choices of camera angle on *Tinder* may reflect the role of partner height in mate selection. It has been often reported that Western women rate taller men as more attractive [43–45], whereas there is no corresponding preference of men for shorter women [46, 47]. Given that *Tinder* arguably emphasizes self-presentation for the purpose of dating, it is plausible that the male bias for camera angles from below stems at least in part from an attempt to appear taller and suggest greater power and physical dominance when posting on this platform in comparison to *Instagram*, which arguably spans a wider range of purposes for self-presentation.

**Table 4. Frequencies of number of dominant poses by sex.**

| Dominance | 4 | 5 | 6 | 7 | 8 | 9 | 10 |
|-----------|---|---|---|---|---|---|----|
| Male | 12 | 21 | 26 | 18 | 11 | 8 | 4 |
| Female | 9 | 29 | 20 | 16 | 17 | 5 | 4 |

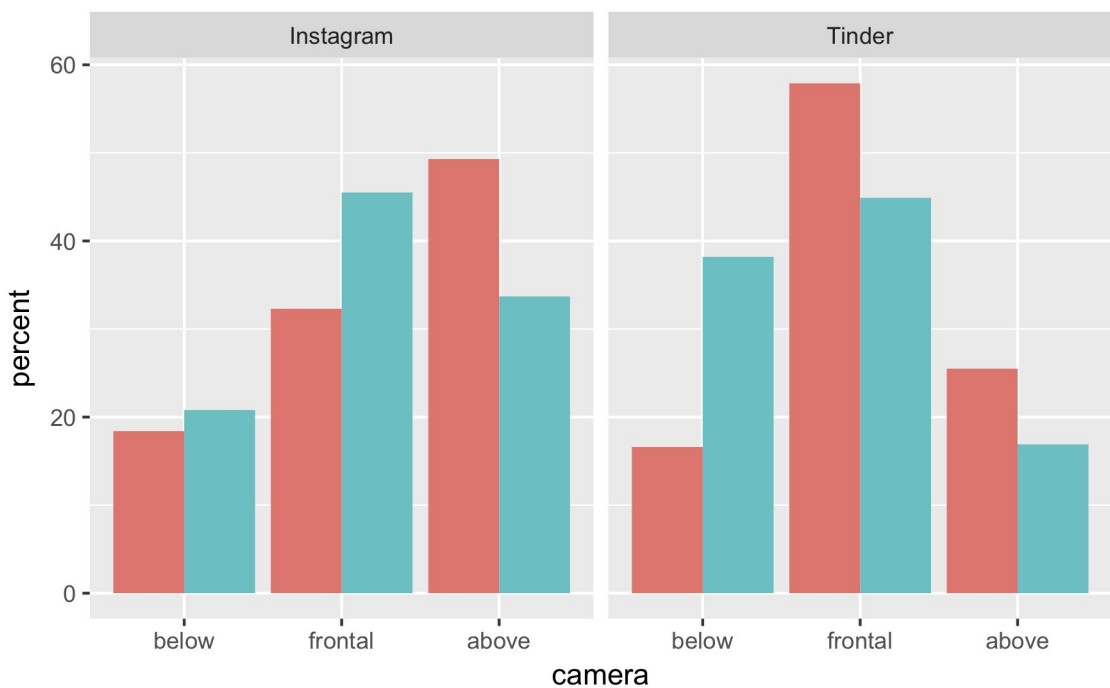

**Fig 1. Percentages of male (pale blue) vs female (pinkish red) selfies by vertical camera angle, on Instagram (current study, left) and on Tinder ([28], right).**

On the other hand, the *Tinder* data of Sedgewick and collaborators also suggest that there may be a female bias for camera angles from above. Given evidence relating female attractiveness and downward head tilt [30], it may be speculated that women are exploiting this nonverbal feature to appear more attractive in a mate selection context. However, our results are not consistent with this interpretation because our *Instagram* sample documented a comparable bias for men takers. Rather, it may be that women have a stronger general preference for more expressive poses than do men. In a communicative context that is relatively unspecific, therefore, they choose the more expressive camera angles from above more than they do neutral frontal angles. Men, conversely, choose neutral frontal angles over the more expressive alternatives. Supporting this interpretation, a recent study of selfies posted on *Instagram* observed that female selfies expressed emotions more strongly than male [22]. In a context explicitly emphasizing mate selection, such as the dating site *Tinder*, female choices of neutral angles instead become more frequent in comparison to cameras above and below, possibly as a consequence of Western social norms that disapprove advertising sexual availability more in women than men.

To sum up, this research extends the findings of Sedgewick, Flath & Elias [28] on the sex differences of the vertical position of the camera when taking selfies. We show that the social platform where the selfies are earmarked makes a notable difference, underling the role of selfies as a form of non-verbal communication. It seems that females tend to prefer selfies from above in all the contexts. Males instead favour selfies from below when communication emphasizes mate selection, whilst they favour frontal selfies otherwise. Given the observational nature of the data we analysed, the current conclusions have the following limitations. The first concerns our claim about the differences in composition between selfies posted on *Tinder* and *Instagram*. While the main purpose of posting on these sites is clearly different, because the comparison we performed is between two different groups of posters it cannot be excluded

in principle that the difference is not due to the communicative intent but to idiosyncratic characteristics, such as personality traits or education level. This may be the case if these characteristics co-vary with frequency of use of these sites. As there is currently no data supporting such covariation, at least to the best of our knowledge, this possibility will need to be tested in further studies. For instance, it should be possible to identify a sample of individuals who post on both sites and perform a within-participant comparison to control for individual differences. Another limitation of our study is that, to test for the consistency of the camera angle, our sample included only selfie takers with at least 10 selfies, but not less. A consequence of this selection criterion is that our sample is representative of the compositional choices of frequent posters but may not capture those of more occasional users. A comparison of occasional and frequent users may indeed be informative, as the former may be assumed to be less aware of the function of selfies for self-presentation and interpersonal communication. Finally, our project focussed generally on attractivity. However, we do not exclude that on social networking sites like Instagram people might also be concerned with looking attractive whilst on Tinder people might also want to convey their sexual availability. Further research will clarify whether the proxemic for attractivity differs from proxemic for sexual availability.

## Author Contributions

**Conceptualization:** Alessandro Soranzo.

**Data curation:** Alessandro Soranzo.

**Formal analysis:** Alessandro Soranzo, Nicola Bruno.

**Methodology:** Alessandro Soranzo.

**Validation:** Alessandro Soranzo.

**Writing – original draft:** Alessandro Soranzo, Nicola Bruno.

**Writing – review & editing:** Alessandro Soranzo, Nicola Bruno.

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
