## [Decision Letter · Decision Letter 0]

12 May 2020

PONE-D-20-08908

Nonverbal communication in selfies posted on Instagram: another look at the effect of gender on vertical camera angle

PLOS ONE

Dear Dr Soranzo,

Thank you for submitting your work to PLOS ONE. I have now secured the reviews of three experts in the field of person perception, particularly in the context of online environments. As you will see from the reviews, all reviewers see merit in the work, but there are some expressions of concern for various points.

Reviewer 1 and Reviewer 3 point out the lack of concrete hypotheses that were made in the paper, and Reviewer 3 draws attention to the comparisons that are made to other data sources (e.g., Tinder) that are relatively unfounded. 

I agree with their assessments, but also agree that the paper is interesting and would make a useful contribution to the literature. As such, I would like to invite you to address the comments of the reviewers and resubmit the manuscript. 

Finally, I hope you, your co-authors, and families are all doing well during the current time.

Regards, 

Alex

We would appreciate receiving your revised manuscript by Jun 26 2020 11:59PM. To enhance the reproducibility of your results, we recommend that if applicable you deposit your laboratory protocols in protocols.io, where a protocol can be assigned its own identifier (DOI) such that it can be cited independently in the future. For instructions see: http://journals.plos.org/plosone/s/submission-guidelines#loc-laboratory-protocols

We look forward to receiving your revised manuscript.

Kind regards,

Alex Jones

Academic Editor

PLOS ONE

Journal Requirements:

2. Please change "female” or "male" to "woman” or "man" as appropriate, when used as a noun.

3. In your Methods section, please include additional information about your dataset and ensure that you have included a statement specifying whether the collection method complied with the terms and conditions for the websites from which you have collected data.

4. Please provide additional details regarding participant consent. In the ethics statement in the Methods and online submission information, please ensure that you have specified (1) whether consent was informed and (2) what type you obtained (for instance, written or verbal, and if verbal, how it was documented and witnessed). If your study included minors, state whether you obtained consent from parents or guardians. If the need for consent was waived by the ethics committee, please include this information.

'The funders had no role in study design, data collection and analysis, decision to publish, or preparation of the manuscript'

Reviewers' comments:

Reviewer's Responses to Questions

**Comments to the Author**

1. Is the manuscript technically sound, and do the data support the conclusions?

Reviewer #1: Yes

Reviewer #2: Yes

Reviewer #3: Yes

2. Has the statistical analysis been performed appropriately and rigorously? 

Reviewer #1: Yes

Reviewer #2: Yes

Reviewer #3: No

3. Have the authors made all data underlying the findings in their manuscript fully available?

Reviewer #1: Yes

Reviewer #2: Yes

Reviewer #3: Yes

4. Is the manuscript presented in an intelligible fashion and written in standard English?

Reviewer #1: Yes

Reviewer #2: Yes

Reviewer #3: Yes

5. Review Comments to the Author

Reviewer #1: The authors analyze selfies posted on Instagram and test whether there are sex differences in camera angle. Camera angle can alter various features of photographs, including the perceived facial appearance of targets. Thus, different camera angles may be employed to emphasize or enhance certain characteristics. A previous study found that on Tinder, men were more likely to post selfies from a low camera angle, whereas women were more likely to post selfies from a high camera angle. In the current study, the authors test whether these findings replicate on a social networking (rather than a dating) website. The study provides novel insights into how men and women differently utilize camera angle when taking selfies. Yet, there are several issues that should be addressed.

Major issues:

1) I would like to see a more detailed explanation of the communicative intentions underlying different camera positions. Which objective features of photos are altered by different vertical camera positions? Which inferences do these differences trigger in perceivers (e.g., Hehman et al., 2013)? And finally, what are the ultimate explanations for sex differences in this practice? In other words, the theoretical depth of the paper could be improved.

2) Whereas the general research questions are explicitly stated at the end of the introduction, the specific hypotheses for the current study were less clear to me. It is true that Instagram is generally used for different purposes than Tinder. However, it is unclear what predictions regarding sex differences in camera positioning would follow from this observation. One the one hand, the effect might only emerge in online dating environments where looking attractive is very important. On the other hand, people might aim to enhance their perceived attractiveness in many situations. Especially on social networking sites like Instagram or Facebook, people might be concerned with looking attractive. These considerations should be discussed. There might also be empirical work on (a) the motives underlying activity on Instagram and (b) how widespread the motive to look attractive is.

3) Most importantly, the main independent variable of interest needs to be specified more clearly. Three features need to be disentangled: camera position, head orientation, and gaze. Did the authors focus only on selfies in which targets looked directly into the camera and heads were vertically angled to face the camera? In this case, camera position would not change the appearance of the face itself. Only the background might change with a higher (lower) camera position leading to more visibility of stimuli below (above) the face. Or did head orientation and gaze also vary? Crucially, this changes the appearance of the face and leads to different trait attributions. For example, people can manipulate their perceived facial width-to-height ratio, which influences various trait attributions (Hehman et al., 2013). This could be achieved (a) by tilting the head up or down while gazing directly into a camera at eye-level (Hehman et al., 2013) or (b) by gazing directly into a camera below eye-level without tilting the head. In short, camera position by itself only changes one, arguably less interesting, aspect of selfies (what is visible in the background). Differences in facial appearance (and resulting inferences), which seem to be the main focus of the current paper, are a function of camera position, head orientation, and gaze.

Minor issues:

4) “It has been proposed (Bruno, Uccelli, Pisu, Belluardo & De Stefani, 2020) that these signals may be conceptualized, within the two dimensional framework of the picture, as analogous to the non-verbal signals that are provided by spatial behaviours in three-dimensional interactions between human as well as non-human animals (Hall, 1966; Hediger,1955; von Uexküll, 1957)” (p.3). This sentence could be clearer, especially as it seems to introduce the theoretical framework of the study. Are the authors proposing that we should study communicative intentions of selfies by focusing on image qualities that can be objectively extracted from the photo’s two-dimensional structure (e.g., face-to-body ratio)? What would be the advantage of this focus compared to competing approaches?

5) Who coded the gender of targets? Were there multiple raters? What was their level of agreement?

6) Tables 3 and 4 could easily be combined in one table.

7) In the General Discussion, the authors compare their results with previous results from Tinder. However, some direct comparisons with Sedgewick’s study, such as “female selfies are much more likely to display a frontal angle on Tinder than they are on Instagram” (p. 8), are not supported with statistical tests. It seems like the authors have the data needed to test for differences between their findings and the findings of the Sedgewick study. These results should be presented in the Results section and not the GD.

8) “Our assessment of interindividual consistency further corroborates this conclusion, suggesting that female and males are similarly consistent in their choices.” (p. 8) Where is this result reported? It looks like the authors only report sex differences in consistent choices of low camera angles, but not sex differences in overall consistency.

9) “In a communicative context that is relatively unspecific, therefore, they choose the more expressive camera angles from above more than they do neutral frontal angles.” In a context explicitly emphasizing mate selection, such as the dating site Tinder, female choices of neutral angles instead become more frequent in comparison to cameras above and below, possibly as a consequence of Western social norms that disapprove advertising sexual availability more in women than men ““ (p. 9) The argument was not clear to me. First, why would the same sexual signal be seen as inappropriate in a mate choice context, but as appropriate in what the authors call a neutral context? If anything, should sexual signaling be seen as less appropriate in a non-mate choice situation? Second, appearing more attractive is different from signaling sexual availability. Is there evidence that camera position influences perceptions of the latter? Finally, even though Tinder and Instagram are designed for different purposes (dating vs. socializing) people might be strongly concerned with their attractiveness in both contexts. Attractiveness may be a strong driver of popularity on Instagram. That is, the comparison between Instagram and Tinder is not a clean comparison of contexts that are characterized by strong vs. weak attractiveness motives. This limitation of the current study should be made explicit.

Bastian Jaeger

References

Hehman, E., Leitner, J. B., & Gaertner, S. L. (2013). Enhancing static facial features increases intimidation. Journal of Experimental Social Psychology, 49(4), 747–754. https://doi.org/10.1016/j.jesp.2013.02.015

Reviewer #2: I loved this paper, great job. As I was reading it I made various notes for myself, including how did you determine gender, why would men be more likely to engage in mate-relevant display on Tinder and women more mate-relevant display on Instagram, and how would you compare the same individuals on the two platforms. But you then covered all these points. I think my only feedback would be that you could have discussed the role of motivation more. Some individual variation in the participants, such as age, would likely impact the motivations to post selfies and therefore the angle the photos are taken, but this wasn't entirely necessary and might needlessly lengthen the manuscript. If you were to lengthen the manuscript, some further discussion on where to go from here would be ideal.

Reviewer #3: The paper provides an investigation into "selfie" angle of inclination based on the gender of the poster while considering the platform utilized (although not directly).

In its current format I consider the paper has limitations which must be addressed in a substantial revision. I will outline below my primary concerns.

Introduction

The rationale for the current study and subsequent hypotheses are not properly developed. In the introduction there is mention of nonverbal behavior, communication intention, mate signaling, and various other concepts which may have intuitive appeal to the question being investigated, yet they are insufficiently articulated and considered. More detail and development is needed, especially for elements of past research and theory the authors consider to most strongly predict the pattern investigated, as well as exclude potential competing approaches. For instance, are the current claims being made about users on such sites in general (gender specific phenomena) or is this a specific sub-group? Given the specific selection criteria, an emphasis should be placed that this relates to individuals that tend to post a specific type of photograph publicly.

There is also little mention of the community's impact on such decisions (i.e. reputation, and feedback). There seems to be an overlap in the rationale presented between this behavior (if present) relating to innate/evolutionary explanations with that generated by social norms or experience. How do we know if such selfies reflect gender differences per se, or if the individual posters are simply emulating "typical" or "successful" behavior (e.g., taking inspiration from popular figures or social norms)? Why were no considerations offered for sexual signaling vs reputation generation (e.g., investigate the pattern of likes for the photos and tendency of the poster to use "gender-congruent angles")?

I was also confused as to the mention of the Tinder data on which no specific analysis was conducted, only an ad hoc comparison. Unless an analysis can be conducted I find any claims to be tentative at best, and should not be placed with such prominence in the Abstract and Introduction.

Methods and Results:

Given the exploratory nature, an a priori power analysis is needed, alongside a justification for the smallest effect size of interest in such research.

The study itself is quite straightforward, yet it seems to be absent of a few metrics which would have drastically improved the data. For instance, was the popularity of the posters considered? The search algorithm on Instagram (when using hashtags) tends to provide a combination of recent posts, most liked posts, most engage posts, and what it considers users may want (based on geography, community, and other nebulous metrics). As such, it is difficult to argue the current sample represents a random set of the Instagram population.

Of the users selected, did the authors verify that there was no community overlap (user-to-user connections?). This may confound the data, as similar groups have more similar public behavior.

I would also consider the chronology of the 10 photos selected to merit investigation (i.e. users may post multiple types of photos initially - exploration behavior - yet based on feedback - likes, reshares, comments - may settle on a specific format). For instance, were the majority of photos that went against your hypotheses posted before or after the ones that matched your hypothesis?

For the analyses presented, I would like to see confidence intervals and effect sizes. Ideally, these would also be presented as Bayesian generalized mixed effects models, given that some of your claims

refer to null effects (page 8). Additionally, the post hoc z-tests for the contingency table must be presented in full (for all cells) and corrections for multiple comparisons should be reported.

I am also uncertain as to the value of the Figure here, given that no direct inferential tests were conducted. A stronger justification is needed.

Discussion

Consider my earlier points I consider the discussion section to need the most attention. A stronger emphasis on the limitations of the current approach is needed. Given that the analysis does not also investigate, for instance, number of likes/reshares/comments for each photo, claims of innate vs social effects are tenuous. The angles of photos may be purely driven by feedback. There is also no clear mention as to the utility of the current findings or how the authors aim to build this research in the future (e.g., laboratory studies investigating if observers judge the same "poster" more/less favorably based on gender-angle congruence; or the impact of attractiveness or other individual characteristics on the propensity to post such photos). Overall, the discussion should focus on what can be inferred from the current data directly, while considering the wider literature and future approaches.

I understand the principle notion behind this research, but more careful consideration must be given to how it is presented, and the implications and limitations of the current data.

6. PLOS authors have the option to publish the peer review history of their article (what does this mean?). If published, this will include your full peer review and any attached files.

Reviewer #1: Yes: Bastian Jaeger

Reviewer #2: Yes: Danielle L Wagstaff

Reviewer #3: No

---

## [Author Response · Author response to Decision Letter 0]

17 Aug 2020

Reviewer #1:

 The authors analyze selfies posted on Instagram and test whether there are sex differences in camera angle. Camera angle can alter various features of photographs, including the perceived facial appearance of targets. Thus, different camera angles may be employed to emphasize or enhance certain characteristics. A previous study found that on Tinder, men were more likely to post selfies from a low camera angle, whereas women were more likely to post selfies from a high camera angle. In the current study, the authors test whether these findings replicate on a social networking (rather than a dating) website. The study provides novel insights into how men and women differently utilize camera angle when taking selfies. Yet, there are several issues that should be addressed.

Major issues:

1) I would like to see a more detailed explanation of the communicative intentions underlying different camera positions. Which objective features of photos are altered by different vertical camera positions? Which inferences do these differences trigger in perceivers (e.g., Hehman et al., 2013)? And finally, what are the ultimate explanations for sex differences in this practice? In other words, the theoretical depth of the paper could be improved.

 Reply: We have expanded the introduction and cited Hehman et al. (see page 3)

2) Whereas the general research questions are explicitly stated at the end of the introduction, the specific hypotheses for the current study were less clear to me. It is true that Instagram is generally used for different purposes than Tinder. However, it is unclear what predictions regarding sex differences in camera positioning would follow from this observation. One the one hand, the effect might only emerge in online dating environments where looking attractive is very important. On the other hand, people might aim to enhance their perceived attractiveness in many situations. Especially on social networking sites like Instagram or Facebook, people might be concerned with looking attractive. These considerations should be discussed. There might also be empirical work on (a) the motives underlying activity on Instagram and (b) how widespread the motive to look attractive is.

 Reply: We have addressed these issues in the conclusions section, where we discuss potential limitations and further directions for research. 

3) Most importantly, the main independent variable of interest needs to be specified more clearly. Three features need to be disentangled: camera position, head orientation, and gaze. Did the authors focus only on selfies in which targets looked directly into the camera and heads were vertically angled to face the camera? In this case, camera position would not change the appearance of the face itself. Only the background might change with a higher (lower) camera position leading to more visibility of stimuli below (above) the face. Or did head orientation and gaze also vary? Crucially, this changes the appearance of the face and leads to different trait attributions. For example, people can manipulate their perceived facial width-to-height ratio, which influences various trait attributions (Hehman et al., 2013). This could be achieved (a) by tilting the head up or down while gazing directly into a camera at eye-level (Hehman et al., 2013) or (b) by gazing directly into a camera below eye-level without tilting the head. In short, camera position by itself only changes one, arguably less interesting, aspect of selfies (what is visible in the background). Differences in facial appearance (and resulting inferences), which seem to be the main focus of the current paper, are a function of camera position, head orientation, and gaze.

 Reply: These issues are tackled in detail in Bruno et al. (2020). We have addressed them in the in the introduction (page 3) and in the discussion (page 9 and 11).

Minor issues:

4) “It has been proposed (Bruno, Uccelli, Pisu, Belluardo & De Stefani, 2020) that these signals may be conceptualized, within the two dimensional framework of the picture, as analogous to the non-verbal signals that are provided by spatial behaviours in three-dimensional interactions between human as well as non-human animals (Hall, 1966; Hediger,1955; von Uexküll, 1957)” (p.3). This sentence could be clearer, especially as it seems to introduce the theoretical framework of the study. Are the authors proposing that we should study communicative intentions of selfies by focusing on image qualities that can be objectively extracted from the photo’s two-dimensional structure (e.g., face-to-body ratio)? What would be the advantage of this focus compared to competing approaches?

 Reply: We agree with the reviewer. In the new version of the manuscript this section has been largely expanded (page 3). 

5) Who coded the gender of targets? Were there multiple raters? What was their level of agreement?

Reply: In the new version of the manuscript we clarified that the gender was coded based on the names posted by subjects (page 5). 

6) Tables 3 and 4 could easily be combined in one table.

 Reply: We understand why the reviewer made this comment. We were not clear enough in specifying that the two tables refer to different analysis. In the new version of the manuscript this is now clarified (page 6).

7) In the General Discussion, the authors compare their results with previous results from Tinder. However, some direct comparisons with Sedgewick’s study, such as “female selfies are much more likely to display a frontal angle on Tinder than they are on Instagram” (p. 8), are not supported with statistical tests. It seems like the authors have the data needed to test for differences between their findings and the findings of the Sedgewick study. These results should be presented in the Results section and not the GD.

 Reply: In the new version of the manuscript, we have added the analysis to compare the two social platforms (page 9).

8) “Our assessment of interindividual consistency further corroborates this conclusion, suggesting that females and males are similarly consistent in their choices.” (p. 8) Where is this result reported? 

It looks like the authors only report sex differences in consistent choices of low camera angles, but not sex differences in overall consistency.

 Reply: In the new version of the manuscript, we have clarified that we measured the overall interindividual consistency and added a Bayes factor 01 supporting our claim (page 8).

9) “In a communicative context that is relatively unspecific, therefore, they choose the more expressive camera angles from above more than they do neutral frontal angles.” In a context explicitly emphasizing mate selection, such as the dating site Tinder, female choices of neutral angles instead become more frequent in comparison to cameras above and below, possibly as a consequence of Western social norms that disapprove advertising sexual availability more in women than men ““ (p. 9) The argument was not clear to me. First, why would the same sexual signal be seen as inappropriate in a mate choice context, but as appropriate in what the authors call a neutral context? If anything, should sexual signaling be seen as less appropriate in a non-mate choice situation? Second, appearing more attractive is different from signaling sexual availability. Is there evidence that camera position influences perceptions of the latter? Finally, even though Tinder and Instagram are designed for different purposes (dating vs. socializing) people might be strongly concerned with their attractiveness in both contexts. Attractiveness may be a strong driver of popularity on Instagram. That is, the comparison between Instagram and Tinder is not a clean comparison of contexts that are characterized by strong vs. weak attractiveness motives. This limitation of the current study should be made explicit.

 Reply: Fair enough. We have made it explicit in the conclusions. 

References

Hehman, E., Leitner, J. B., & Gaertner, S. L. (2013). Enhancing static facial features increases intimidation. Journal of Experimental Social Psychology, 49(4), 747–754. https://doi.org/10.1016/j.jesp.2013.02.015

Reviewer #2:

 I loved this paper, great job. As I was reading it I made various notes for myself, including how did you determine gender, why would men be more likely to engage in mate-relevant display on Tinder and women more mate-relevant display on Instagram, and how would you compare the same individuals on the two platforms. But you then covered all these points. I think my only feedback would be that you could have discussed the role of motivation more. Some individual variation in the participants, such as age, would likely impact the motivations to post selfies and therefore the angle the photos are taken, but this wasn't entirely necessary and might needlessly lengthen the manuscript. If you were to lengthen the manuscript, some further discussion on where to go from here would be ideal.

 Reply: Thank you for your positive evaluation. The issue of motivation was raised also by reviewer 1. We have improved the introduction accordingly. Please refer to our response to reviewer 1 for details.

Reviewer #3: 

The paper provides an investigation into "selfie" angle of inclination based on the gender of the poster while considering the platform utilized (although not directly).

In its current format I consider the paper has limitations which must be addressed in a substantial revision. I will outline below my primary concerns.

Introduction

The rationale for the current study and subsequent hypotheses are not properly developed. In the introduction there is mention of nonverbal behavior, communication intention, mate signaling, and various other concepts which may have intuitive appeal to the question being investigated, yet they are insufficiently articulated and considered. More detail and development is needed, especially for elements of past research and theory the authors consider to most strongly predict the pattern investigated, as well as exclude potential competing approaches. For instance, are the current claims being made about users on such sites in general (gender specific phenomena) or is this a specific sub-group? Given the specific selection criteria, an emphasis should be placed that this relates to individuals that tend to post a specific type of photograph publicly.

 Reply: Yes of course. The paper is about selfie taking and sharing as a novel form of non-verbal communication. It applies to individuals that use this mode of communication. 

There is also little mention of the community's impact on such decisions (i.e. reputation, and feedback). There seems to be an overlap in the ration presented between this behavior (if present) relating to innate/evolutionary explanations with that generated by social norms or experience. How do we know if such selfies reflect gender differences per se, or if the individual posters are simply emulating "typical" or "successful" behavior (e.g., taking inspiration from popular figures or social norms)? Why were no considerations offered for sexual signaling vs reputation generation (e.g., investigate the pattern of likes for the photos and tendency of the poster to use "gender-congruent angles")?

 Reply: These are valid concerns and we have addressed them in the conclusion of the paper, where we discuss current limitations and further directions for study. Our aim here was to perform an initial investigation of the hypothesis that the composition of selfies may reflect communicative intention, as implied by the intended platform for posting. In an observational study such as this, some limitations in the scope of the conclusions are to be expected.

I was also confused as to the mention of the Tinder data on which no specific analysis was conducted, only an ad hoc comparison. Unless an analysis can be conducted I find any claims to be tentative at best, and should not be placed with such prominence in the Abstract and Introduction.

 Reply: In the new version of the manuscript, we have added the analysis to compare the two social platforms (page 8).

Methods and Results:

Given the exploratory nature, an a priori power analysis is needed, alongside a justification for the smallest effect size of interest in such research.

Reply: In the new version of the manuscript the sample size is justified on page 5.

The study itself is quite straightforward, yet it seems to be absent of a few metrics which would have drastically improved the data. For instance, was the popularity of the posters considered? The search algorithm on Instagram (when using hashtags) tends to provide a combination of recent posts, most liked posts, most engage posts, and what it considers users may want (based on geography, community, and other nebulous metrics). As such, it is difficult to argue the current sample represents a random set of the Instagram population.

Of the users selected, did the authors verify that there was no community overlap (user-to-user connections?). This may confound the data, as similar groups have more similar public behavior.

I would also consider the chronology of the 10 photos selected to merit investigation (i.e. users may post multiple types of photos initially - exploration behavior - yet based on feedback - likes, reshares, comments - may settle on a specific format). For instance, were the majority of photos that went against your hypotheses posted before or after the ones that matched your hypothesis?

Reply: We agree that possible biases in sampling need to be considered. We have discussed this issue in the conclusions, where we address potential limitations of the current conclusions. We remain sceptical however that popularity metrics (which indeed tend to be nebulous, as you also acknowledge) would help much in this respect. On the other hand, your suggestion about looking into the chronology of the posts is very interesting and we have decided to include it as a further direction for study. We are in fact deeply interested in the role of exploratory behaviours in the form of nonverbal communication instantiated by selfies, a notion which is developed in depth in a much longer recent paper from our group which formed the theoretical starting point for this work (Bruno, Uccelli, Pisu, Belluardo & De Stefani, 2020, frontiers in Human-Media Interaction). This analysis however would require setting up a new study, and obtaining and implementing the appropriate ethics-approval procedures as it would raise privacy concerns related to recording personal data from the online profiles. In our study, we deliberately discarded all this information precisely to avoid having to deal with these delicate issues. 

For the analyses presented, I would like to see confidence intervals and effect sizes. Ideally, these would also be presented as Bayesian generalized mixed effects models, given that some of your claims refer to null effects (page 8). Additionally, the post hoc z-tests for the contingency table must be presented in full (for all cells) and corrections for multiple comparisons should be reported.

 Reply: In the new version of the manuscript we have added the CI and effect sizes. The analysis to support the claim based on null effects was replaced with a Bayesian test (page 8). 

I am also uncertain as to the value of the Figure here, given that no direct inferential tests were conducted. A stronger justification is needed.

 Reply: We agree with the reviewer. In the new version of the manuscript (page 8) we have added the inferential test comparing the two social networks, making the figure meaningful. 

Discussion

Consider my earlier points I consider the discussion section to need the most attention. A stronger emphasis on the limitations of the current approach is needed. Given that the analysis does not also investigate, for instance, number of likes/reshares/comments for each photo, claims of innate vs social effects are tenuous. The angles of photos may be purely driven by feedback. There is also no clear mention as to the utility of the current findings or how the authors aim to build this research in the future (e.g., laboratory studies investigating if observers judge the same "poster" more/less favorably based on gender-angle congruence; or the impact of attractiveness or other individual characteristics on the propensity to post such photos). Overall, the discussion should focus on what can be inferred from the current data directly, while considering the wider literature and future approaches. I understand the principle notion behind this research, but more careful consideration must be given to how it is presented, and the implications and limitations of the current data.

 Reply: Fair enough. We have rewritten the discussion section to address these concerns.

---

## [Editor Report · Decision Letter 1]

20 Aug 2020

Nonverbal communication in selfies posted on Instagram: another look at the effect of gender on vertical camera angle

PONE-D-20-08908R1

Dear Dr. Soranzo,

We’re pleased to inform you that your manuscript has been judged scientifically suitable for publication and will be formally accepted for publication once it meets all outstanding technical requirements.

Kind regards,

Alex Jones

Academic Editor

PLOS ONE
---

## [Editor Report · Acceptance letter]

24 Aug 2020

PONE-D-20-08908R1 

Nonverbal communication in selfies posted on *Instagram*: another look at the effect of gender on vertical camera angle 

Dear Dr. Soranzo:

I'm pleased to inform you that your manuscript has been deemed suitable for publication in PLOS ONE. Congratulations! Your manuscript is now with our production department. 

Kind regards, 

on behalf of

Dr. Alex Jones 

Academic Editor

PLOS ONE